# Detection of Intoxicated Passengers at Stations to Prevent Accidents on Railway Platforms

Naoki Kikitsu
Dept. of Advanced Electronic Engineering,
School of Engineering,
Shibaura Institute of Technology
Tokyo, Japan
Email:ag20059@shibaura-it.ac.jp

Chinthaka Premachandra
Dept. of Advanced Electronic Engineering,
School of   Engineering,
Shibaura Institute of Technology
Tokyo, Japan
Email:chintaka@shibaura-it.ac.jp

*Abstract*— **Accidents due to falls from station platforms occur frequently, with more than half caused by intoxicated passengers. However, effective preventive measures, such as the installation of platform doors, are expensive and difficult to implement in small and medium-sized stations. As a new preventive measure, this study proposes a low-cost wobble detection system that requires only a camera and a lightweight computer. In the proposed method, wobbliness is detected by evaluating the pattern of fluctuations in the skeleton coordinates over a specific time frame. We conducted various experiments to evaluate the proposed wobble detection approach in simulated station situations. By assessing the presence or absence of wobble at three locations (hip, knee, and ankle), we found the accuracy rates to be 95.0% for the hip, 93.3% for the knee, and 91.7% for the ankle, demonstrating high accuracy.**

*Keywords*— **detection of intoxicated passengers, railway platform accidents, skeleton coordinates, wobble detection**

## I. INTRODUCTION (*HEADING 1*)

Nowadays, accidents involving a train colliding with another train due to a fall from a station platform onto the railway tracks occur frequently worldwide. According to a report by the Ministry of Land, Infrastructure, Transport and Tourism in 2028, there were 2,887 accidents involving falls from station platforms, of which 160 resulted in collisions with trains. Analyzing the causes of these falls reveals that more than half (about 57%) were caused by intoxicated passengers. In other words, passengers who become unsteady due to alcohol consumption and fall onto the platform are a major factor in these accidents [1][2][3].

Various preventive measures have been implemented to address this issue. For example, platform doors are installed, station staff are stationed on the platform, and cameras monitor the area. Data on platform doors indicate that although Japan has about 10,000 stations, only around 1,000 have platform doors, accounting for only about 10% of all stations[4]. The primary reason for this low percentage is the high cost of installation. The initial cost of installing platform doors ranges from several hundred million to several billion yen per station (including platform reinforcement), and the annual maintenance and management cost is around 3 to 20 million yen per line, along with other significant renewal costs. Consequently, it is challenging to implement platform doors at all stations, especially small to medium-sized ones where platform width may be too narrow.

Given these challenges, it is crucial to consider alternative fall-prevention measures that do not rely on platform doors. Additionally, relying on station staff and camera monitoring can lead to missed detections of intoxicated individuals, as these methods depend on human observation[5]. On the other hand, some passenger privacy-secured systems have been developed to detect unsafe activities, such as smartphone usage on platforms, using depth cameras [6]. However, detecting wobble with depth information can be challenging because the movements of other pedestrians can also interfere with depth data. Some systems for detecting wobble situations are addressed in the literature, but pedestrian privacy protections are not well considered [7]. In this paper, we use only the variations of hip, knee, and ankle coordinates. Thus, this method can be applied while protecting pedestrian privacy by capturing only the lower body parts of pedestrians with a general camera. The proposed method can be implemented at a relatively low cost at small and medium-sized stations.

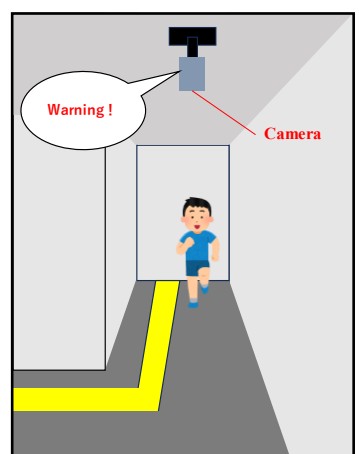

Fig. 1. Assumed diagram of the proposed system

As shown in Fig. 1, the system can detect the presence of the device in advance at the safe passageway from the ticket gate to the platform. We believe that this type of system can be installed at multiple locations within a station, thereby contributing to the overall safety of the station. Given the potential of this system, we believe it can be applied not only to station platforms but also to other hazardous areas, such as public facilities and sidewalks, where many accidents involving falls occur.

## II. OVERALL FLOW OF WOBBLE DETECTION

The overall flow of the wobble detection method proposed in this study is illustrated in Fig. 2. This method discriminates between normal gait and staggering gait based on the characteristics of the generated trajectory to detect wobbling. In this study, we use OpenPose, a computer vision library that employs deep learning to estimate the positions of a person's skeleton. The skeleton data from OpenPose are then graphed to visualize the trajectories of the skeletons, and the features of these trajectories are used to determine whether the individual has a normal or staggering gait.

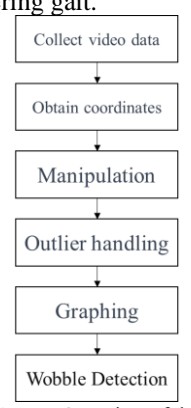

Fig. 2. System Overview of this Study

### A. Video Data Capturing

In this study, two patterns of walking—normal walking and staggering walking—were prepared. Since the focus is on pedestrians passing through a ticket gate at a train station, the primary target is a pedestrian approaching the camera. To simulate this situation, we installed the camera at a high location, such as in a corridor, and recorded videos of pedestrians coming toward the camera. The angle of view during video recording is illustrated in Fig. 3. As shown in Fig. 3, the subject walked towards the camera, which was positioned at a height of about 1.8 meters. A standard 2D camera was used, installed to capture a wide area of the corridor.

### B. Skeleton Coordinate Acquisition by OpenPose

In this study, OpenPose is used to obtain coordinate data from the captured videos. OpenPose is open-source software for estimating human posture and motion. It detects human joints and body parts and estimates their location information. OpenPose uses a machine learning-based approach for posture estimation, aiming to detect key points (joints and body parts) of the human body from images and videos, and to connect these key points to estimate the structure and movements of the human body.

Identify applicable funding agency here. If none, delete this text box.

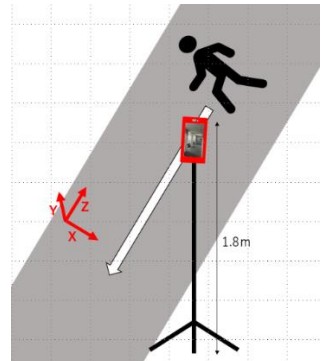

Fig. 3. Camera Installation and Filming Methods

OpenPose performs the process in the following steps:

- Image Feature Extraction: A feature map is generated from the input image using a CNN. This feature map abstracts pixel-level information and stores the extracted features through convolutional and pooling layers.

- Estimation of Key Points: The generated feature maps are used to estimate the key points of the human body in the image. A set of candidates representing the locations of joints and body parts is generated.

- Key Point Merging and Posture Estimation: The candidate key points are connected to form the human body structure. In this step, the final posture is estimated by considering the relationships among the key points and the context.

- Interpretation and Analysis of Posture: The final posture estimation is analyzed, and algorithms and methods are applied to detect anomalies such as a wobbling gait.

Thus, OpenPose can estimate the posture information of a human body from images through complex processing. OpenPose is used in various situations and has been referenced in many studies cited in this research. Fig. 4 shows how the joint positions were actually estimated in this study.

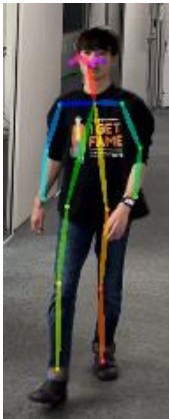

Fig. 4. Joint Estimation with OpenPose

## C. Removal of Null-data from OpenPose

In this study, several processes are applied to the raw coordinate data acquired by OpenPose[8][9]. First, the coordinates of each region are combined into a single point to simplify data handling. The joint coordinates obtained for the left and right joints of one region are merged into the coordinates of an intermediate point, resulting in the coordinate data for three locations. For example, the coordinates of the hip joint are determined as the midpoint between the coordinates of the right hip joint and the left hip joint. Next, as shown in Fig. 3, since the pedestrian is assumed to be approaching the camera, we judged that the transition of the coordinates in the Y direction is not directly related to the determination of wobbliness. Therefore, we focused only on the transition of coordinates in the X direction. Thus, we use three sets of data—the trajectory of the X coordinates of the hip, the X coordinates of the knee, and the X coordinates of the ankle—to determine whether wobbliness is present. Additionally, in the coordinate data obtained by OpenPose, there were frames at the beginning and end of the video where the person was not recognized, resulting in empty data where the coordinates were not properly acquired. That data were complemented by taking the average of the values before and after the corresponding empty data and replacing them with the corresponding empty data following the equation (1). Let $x[j]$ denote the relevant empty data, and $x[j-1], x[j+1]$ denote the data in the frames before and after $x[j]$, as shown in the equation (1).

$$x[j] = \frac{(x[j-1] + x[j+1])}{2} \qquad (1)$$

This process resulted in a list of coordinate data with values for considered frames. However, data that could not be complemented were removed from the analysis.

## D. Data Visualization by Graphing

To visualize the transition of the list of coordinate data for normal gait and wobbly gait, we generated graphs using the data from OpenPose after applying the complementing process. As an example, Fig. 5 shows graphs of hip joint coordinates for two patterns: (a) data of normal gait and (b) data of staggering gait.

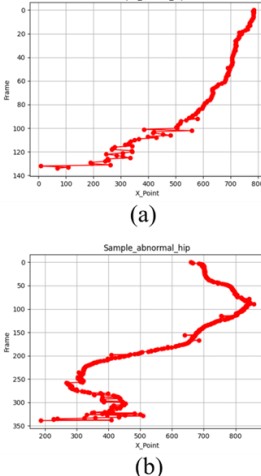

(a)

(b)

Fig. 5. Graph coordinate data

## E. Outlier Removal

As shown in the graph in Fig. 7, the data in this state contains many abnormal values due to minor blurring and false positives, making it difficult to discern accurate trends. The following two methods are used to remove these outliers. First, outlier processing is performed using a statistical measure called the Z-score. The Z-score indicates how far a particular data point is from the mean, considering the mean and standard deviation of the entire data set. The Z-score is calculated using the following equation (2).

$$Z = \frac{(X - \mu)}{\sigma} \qquad (2)$$

where X represents the individual data points, µ is the mean of the data set, and σ is the standard deviation of the data set. When Z is 1, it means that the point is one standard deviation away from the mean. In this case, data points with a Z-score greater than 2 are excluded as outliers. Outliers still existed even after processing using the Z-score. This is because outliers are estimated based on variance (standard deviation), and depending on the skewness of the values, not all outliers may be fully detected. Therefore, additional processing was applied to address this issue. We examined the consecutive coordinate data points one by one, and if the difference between the current data and the previous data was greater than a certain threshold value, the current data was excluded as an outlier. Specifically, if the X-coordinate differed from the previous data by 30 or more, the data was considered an outlier and excluded. By applying this dual-stage outlier removal process, we were able to refine the coordinate, making it more accurate and smooth. The graphs of hip joint coordinates for the two patterns (a: normal gait data and b: wobbly gait data) after outlier processing, as shown in Fig. 5, are presented in Fig. 6.

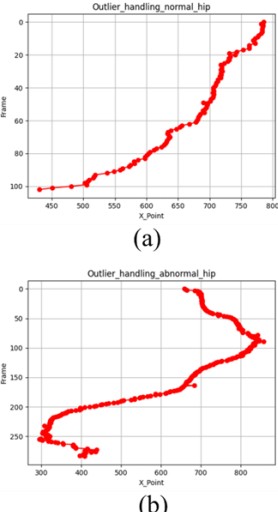

(a)

(b)

Fig. 6. Coordinate data after outlier removal processing

## F. Wobble Walking Detection

In this study, wobble was detected by evaluating the pattern of fluctuations in the average value over a specific time frame. The data set was divided into an arbitrary number of segments (five frames in this case), and the average value within each

segment was calculated. The number of times the calculated value increased or decreased from the previous value was then counted. As observed from Fig. 5, the X-coordinate tends to change monotonically in the data of normal walking, whereas it tends to fluctuate repeatedly in the data of staggering walking. To quantify this, the rate of increase (rate) was calculated using the following equation (3), where the number of increases is defined as "inc" and the number of decreases as "dec".

$$rate = 100 \frac{inc}{inc + dec} \ [\%] \tag{3}$$

If the rate of increase (rate) exceeds 80% or less than 20%, the data is judged as "monotonically increasing or monotonically decreasing" and is classified as normal gait data. On the other hand, if the rate is between 20% and 80%, the data is considered neither monotonically increasing nor monotonically decreasing, and is classified as a staggering gait.

## III. VERIFICATION EXPERIMENT

### A. Experimental Environment

As a validation experiment, we prepared 30 videos of normal walking and 30 videos of wobbling walking to evaluate the developed wobble detection system. In the experiment, we applied the proposed method to these 60 videos and determined the occurrence of wobble based on the skeleton positional information at the hip, knee, and ankle points. We evaluated the performance of the proposal using the metrices present below. As indices of the results, we present the Accuracy, Precision, and Recall. Each of them is discussed in detail based on the following four quadrants.

- TP(True Positive) : Judging the wobble as "abnormal" walking.
- TN(True Negative) : Normal walking is judged as "normal".
- FP(False Positive) : Normal walking is judged as "abnormal"(wobbles)
- FN(False Negative) : Judging the wobble as "normal" walking.

The following is an explanation of the result indices along with their formulas.

Accuracy refers to the percentage of correct judgments among the entire data set. It is expressed by the following equation (4).

$$Accuracy = \frac{TP + TN}{TP + FP + TN + FN} \tag{4}$$

Precision refers to the percentage of correct answers among those judged as abnormal. As precision increases, it means there are fewer false positives. It is expressed by the following equation (5).

$$Precision = \frac{TP}{TP + FP} \tag{5}$$

Recall refers to the percentage of correct answers among all actual cases of wobbling motion. It is expressed by the following equation (6).

$$Recall = \frac{TP}{TP + FN} \tag{6}$$

### B. Experimental Results

In this experiment, we evaluated the coordinate data of the hip, knee, and ankle from a total of 60 videos (30 videos of normal walking and 30 videos of staggering walking). First, Table 1 shows the hip coordinate-based judgment.

TABLE I.   RESULTS OF HIP JOINT BASED DETERMINATION

| Hip coordinate-based judgment | | Prediction | |
|---|---|---|---|
| | | Abnormal | Normal |
| True | Abnormal | 27 | 3 |
| | Normal | 0 | 30 |

Next, Table 2 shows the results of the knee coordinate-based judgment.

TABLE II.   RESULTS OF KEE BASED DETERMINATION

| Knee coordinate-based judgment | | Prediction | |
|---|---|---|---|
| | | Abnormal | Normal |
| True | Abnormal | 26 | 4 |
| | Normal | 0 | 30 |

Finally, Table 3 shows the results for the ankle coordinate-based judgment.

TABLE III.   RESULTS OF ANKLE JOINT BASED DETERMINATION

| Ankle coordinate-based judgment | | Prediction | |
|---|---|---|---|
| | | Abnormal | Normal |
| True | Abnormal | 25 | 5 |
| | Normal | 0 | 30 |

Table 4 shows a summary of the calculated index values.

TABLE IV.    RESULTS DETERMINATION RESULTS

| Considered body coordinate | Accuracy [%] | Precision [%] | Recall [%] |
|---|---|---|---|
| Hip | 95.0 | 100.0 | 90.0 |
| Knee | 93.3 | 100.0 | 86.7 |
| Ankle | 91.7 | 100.0 | 83.3 |

According to the results, we found the accuracy rates to be 95.0% for the hip, 93.3% for the knee, and 91.7% for the ankle, demonstrating high accuracy. The calculation results show that the precision is 100% for all cases, indicating that there are no false positives. In other words, no non-wobble cases were detected as wobble. The recall is 90.0% for the hip, 86.7% for the knee, and 83.3% for the ankle, indicating that in some cases wobbles are not correctly detected.

## IV.    CONCLUSIONS

In this study, we proposed a wobble detection method using a camera as one of the measures to prevent accidents involving falls from station platforms. We developed a method that uses the movement of coordinates of a pedestrian's lower body parts, such as the hip, knee, and ankle, to determine whether the pedestrian is wobbling. According to the experimental results, the proposed method was able to detect wobble with a high percentage of correct responses for each joint, with the hip joint being the most accurate.

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
