# OpenReview forum: "Detection of Intoxicated Passengers at Stations to Prevent Accidents on Railway Platforms"
_IEEE.org/ICIST/2024/Conference — IEEE ICIST 2024 Conference Submission_

### Official Review · Reviewer_qeAh · 2024-08-21
**This  study proposes a low-cost wobble detection system that requires  only a camera and a lightweight computer. In the proposed  method, wobbliness is detected by evaluating the pattern of  fluctuations in the skeleton coordinates over a specific time frame. The idea is interesting and the obtained result is valuable. This paper may be accepted if the following problems can be clarified.**

**Rating:** 7
**Confidence:** 3

**Review:**

1. The references should be updated. Some closely related and new references should be added to show to further explain the novelty and innovation of the work.
2. The disadvantages of the proposed method and the direction of the next work should be explained in the conclusion.
3. It can be compared with existing articles to make the innovation point clearer.

---

### Official Review · Reviewer_hwPu · 2024-08-21
**accept**

**Rating:** 7
**Confidence:** 3

**Review:**

Comment:
This paper considered a wobble detection method using a camera as one of the measures to prevent accidents involving falls from station platforms. The theory is correct and can be accepted after responding the following comments.
(1)	In the introduction, it is not enough to state the current work. It should be expended and reconstructed.
(2)	There are many typos and grammar errors. The authors should have a native English speaker or software packages to perform the editing check.
(3)	The format of the references needs to further adjustment.

---

### Official Review · Reviewer_TW9w · 2024-08-22
**this work is well organized and appears potentially interesting, it can be accepted with a little modification.**

**Rating:** 8
**Confidence:** 4

**Review:**

This study proposes a low-cost wobble detection system that requires only a camera and a lightweight computer. The openPose is used to obtain coordinate data from the captured videos. A feature map is generated from the input image using a CNN. This feature map abstracts pixel-level information and stores the extracted features through convolutional and pooling layers. In general, this work is well organized and appears potentially interesting, it can be accepted with a little modification.
1.	Please provide a detailed analysis of the simulation results.
2.	Please explain what the innovative points of this system are compared to other systems.
3.	Please explain the derivation of Equation (1).
4.	What is the future direction of the research described in this article?

---

### Decision · Program_Chairs · 2024-09-06

Accept (Oral)